# Ecological Risk Assessment and Source Apportionment of Heavy Metals in the Soil of an Opencast Mine in Xinjiang

**DOI:** 10.3390/ijerph192315522

**Published:** 2022-11-23

**Authors:** Tingyu Fan, Jinhong Pan, Xingming Wang, Shun Wang, Akang Lu

**Affiliations:** 1School of Earth and Environment, Anhui University of Science and Technology, Huainan 232001, China; 2Institute of Environmental Friendly Materials and Occupational Health, Anhui University of Science and Technology, Wuhu 241002, China; 3Anhui Engineering Laboratory for Comprehensive Utilization of Water and Soil Resources & Ecological Protection in Mining Area with High Groundwater Level, Huainan 232001, China

**Keywords:** heavy metals, potential ecological risks, health risks, source analysis

## Abstract

To study the influence of open-pit coal mining on the surrounding soil environment and human health, this study selected the Hongshaquan coal mine in Xinjiang as the research area and took 31 soil samples from the dump and artificial forest of the mining area. The contents of seven heavy metals (As, Cd, Cr, Cu, Ni, Pb and Zn) in the soil were analyzed. The pollution index method, geoaccumulation index method (I_geo_), potential ecological risk index method, health ecological risk assessment model and principal component analysis (PCA) were used to evaluate and analyze the heavy metal pollution, potential ecological risk and health ecological risk of the soil. The results showed that compared with the background value of soil in Xinjiang, except for Pb, other heavy metal elements were essentially pollution-free and belonged to the low ecological risk area. The health risk assessment model showed that Pb and As were the main pollution factors of noncarcinogenic risk, and that exposure to Ni, Pb and As had a lower carcinogenic risk. The PCA showed that Cu, Cr, Ni, Pb, As and Zn in the dump were from transportation and industrial activities, Cd was from natural resources, and Cr, Zn, Ni, Cd and Pb were from transportation in the artificial forest. Cu came from industrial sources and As from soil parent material. The dump was more seriously disturbed by human factors than by artificial forests. Our research provides a reference for heavy metal pollution and source analysis caused by mining.

## 1. Introduction

Coal resources are one of the main energy bases in China. Coal mining is closely related to the environment. Currently, soil heavy metal pollution has become a serious environmental problem [1,2]. China’s soil heavy metal overlimit rate was reported to be 16.1% in 2014 [3]. In the 14th Five-Year Plan released in 2020 [4], China formulated a plan for mine ecological restoration and carried out verification of mines left over by history. It further strengthened the ecological restoration of key watersheds and regional historical mines. Xinjiang is an important area for coal resource distribution and mainly an open-pit coal mine [5]. Open-pit mining is an efficient and economical mode of mineral resource development, which can produce a variety of buried substances (metal, non-metallic and rare semiconductor materials, etc. [6,7,8,9]). But it leads to a series of problems, such as soil heavy metal pollution, land degradation and intractable carbon dioxide emissions by eliminating vegetation and permanently altering topography, soil and subsurface geological structures [10,11,12,13].

Heavy metal pollution in soil is closely related to human activities during mining in open-pit mines [14,15,16]. For example, heavy metals spread far and wide through dust and wind effects from various human activities, migrate to nearby water bodies and soils, and destroy the local ecological environment [17,18]. They subsequently enter the human body in various ways, such as plant absorption, human skin contact and inhalation [19,20]. Although some heavy metals (e.g., Cu, Zn) are trace elements required for biological growth, long-term exposure to heavy metals may pose a variety of potential risks to humans. For example, Cd and Cr can have carcinogenic effects [21,22], Hg and Cu can damage the nervous system, and As can cause skin and liver diseases [23,24,25]. Due to the toxicity, persistence, resistance to degradation and ability of many heavy metals to enter the food chain, soil heavy metal pollution has become a focus of attention [26,27,28].

Currently, there are many methods for evaluating heavy metals in soil, such as the pollution index method based on the environmental background value, the Nemerow comprehensive pollution index method and the geoaccumulation index method. The potential ecological risk index method comprehensively considers various potential factors, the assessment model method for evaluating human health risks, and the principal component analysis (PCA) method and positive matrix factorization (PMF) method for source analysis [29,30,31,32]. We studied the difference in heavy metal pollution in different functional areas of open pit mines and its harm to human health, so as to better control the heavy metal pollution in open pit mines. We chose some methods according to the actual situation to analyze the heavy metal pollution in open pit mines. In this study, the dump and artificial forest in the Hongshaquan mining area of Xinjiang were taken as the research area, and the soil samples were collected and tested to analyze the heavy metal concentration and physical and chemical properties of the soil in the mining area. The pollution index method, geoaccumulation index method and potential ecological risk were used to evaluate the pollution of heavy metals, and the health and ecological risk assessment model was then used to determine the health and ecological risk of the mining area. Finally, the principal component analysis method was used to analyze the sources of heavy metals. This study provides a reference for the comprehensive management and ecological restoration of mining areas.

## 2. Methods

### 2.1. Study Area

The Xinjiang Hongshaquan coal mine belongs to Qitai County, located in the southeastern Junggar Basin, in the Gobi Desert. The geographical coordinates are E 90°15′40″, N 44°40′42″. Most parts of the mining area have a perennial no-surface water flow, are perennially windy, mostly northwest wind-based, which results in serious dust. The highest temperature in summer is 43.2 °C, the winters are dry, and there is a large temperature difference between day and night; the annual evaporation capacity is of 1200~22,350 mm. The study area belongs to the Xinjiang Desert vegetation area, and the vegetation coverage is low: less than 10%.

After a field investigation, it was established that the main natural vegetation in the mining area is *Haloxylon ammodendron*, *Nitraria tangutorum*, *Salsola foliosa*, *Ceratocarpus arenarius*, *Anabasis salsa* and other xerophytic and hyperxeric small semi-shrubs and semi-shrubs. In order to improve the environment as well as the ecological environment, the mine attempted to plant trees in different regions to form artificial forests. The soil stripped from the open pit mining pit was stacked on the dump, forming three platforms with different altitudes (706 m, 726 m, 746 m). In order to fully grasp the heavy metal system on the surface of the study area, sampling points were selected in two areas of artificial forest and dump.

### 2.2. Soil Sample Collection and Analysis

According to the field survey, in July 2020, two research areas of dump and plantation were determined, and a total of 31 samples were collected at a depth of 0–20 cm on the soil surface using a soil-layered sampler (Figure 1). There were three platforms with different heights in the dump area, and two sampling lines (No.1 and No.2) were arranged. Each sampling line had nine points, and each platform had three points with a spacing of 50 m. A sampling line (No.3 and No.4) was laid on the left and right sides of the road in the plantation area. There were six points on the No.3 sampling line and seven points on the No.4 sampling line, with a spacing of 50 m. The sampling records are shown in Table 1. The samples were stored in self-sealing bags after collection, which was convenient for the analysis and detection of laboratory physical and chemical properties.

The samples were air-dried and crushed in the laboratory, filtered through a 63 μm sieve and then digested. The specific process was as follows: a 0.1 g sample was placed in a 100 mL conical flask, mixed with 6 mL aqua regia, placed in a glass funnel, heated and digested for 2 h. Then the contents of seven heavy metals were determined by inductively coupled plasma-mass spectrometry (ICP-MS). Three sets of parallel tests were conducted, and the concentration of heavy metals was represented as the average of the repeating samples.

### 2.3. Pollution Index Method and Geoaccumulation Index Method

First, the pollution index was used to evaluate the degree of soil heavy metal pollution [33,34]. CF (pollution factor) represents the ratio of the measured value (Cn) of soil heavy metal elements to the background value (Cb) of soil heavy metal elements in Xinjiang. The modified pollution factor (MCF) is the Hakanson pollution index, which combines the single pollution evaluation index of each soil heavy metal sample [35]. The calculation formulas of CF and MCF are as follows.
(1)$$\mathrm{CF}=\frac{\mathrm{Cn}}{\mathrm{Cb}}$$
(2)$$\mathrm{MCF}=\frac{\sum_{i=1}^{n}\mathrm{CF}}{n}$$

CF is single factor pollution, and Cn and Cb are the concentration and background value of elements in Xinjiang soil. N is the number of heavy metals. When CF is greater than 1, it indicates element enrichment; when CF is less than 1, it represents loss. MCF is classified in Appendix A.

The geoaccumulation index method is a widely used evaluation method for heavy metals in soil [36]. The calculation formula is as follows.
(3)$$I_{\mathrm{geo}}=\log_{2}\left[\frac{\mathrm{Cn}}{\left(K\times \mathrm{Bn}\right)}\right]$$

Cn represents the measured value of soil heavy metals (g/kg), Bn is the geochemical background value of the element (g/kg), and K is a coefficient considering the variation of the background value caused by rock differences at different locations (in this study, K = 1.5). Its classification is shown in Appendix A.

### 2.4. Potential Ecological Risk Index Method

Potential ecological risk assessment using the potential ecological risk index method is one of the most common and widely used methods to evaluate the degree of heavy metal pollution [37]. Compared with other methods, this method considers the synergistic effect of various heavy metal elements, the toxicity level of heavy metals, the concentration of pollutants and the sensitivity of ecology to heavy metals. It has a certain ecological rationality. The calculation formula is as follows.
(4)$$C_{f}^{i}=\frac{C^{i}}{C_{n}^{i}}$$
(5)$$E_{r}^{i}=T_{r}^{i}\times C_{f}^{i}$$
(6)$$\mathrm{RI}=\sum_{i}^{m}E_{r}^{i}=\sum_{i}^{m}T_{r}^{i}\times \frac{C^{i}}{C_{n}^{i}}$$
where $C_{f}^{i}$ is the pollution coefficient of the first heavy metal; $C^{i}$ is the measured value of the ith heavy metal content in the sample, mg·kg^−1^; $C_{n}^{i}$ is the background value of the ith heavy metal in the soil (Cd = 0.12, Pb = 13.5, Cu = 35.8, Zn = 68.8, Ni = 26.4, Cr = 39.6, As = 9.09), mg·kg^−1^; $E_{r}^{i}$ is the potential ecological risk coefficient of the ith heavy metal in the soil; $T_{r}^{i}$ is the toxicity coefficient of the ith heavy metal in the soil (Appendix A); and RI is the potential ecological risk index of various heavy metal elements. $E_{r}^{i}$ and RI can evaluate the potential ecological risk degree of a certain pollutant and many kinds of pollutants, respectively. The grading standards of each index are shown in Appendix A.

### 2.5. Health Risk Assessment

A health risk assessment model developed by the US Environmental Protection Agency was used to assess the noncarcinogenic and carcinogenic effects of heavy metals on the human body. Considering the differences in vulnerability among different age groups, this study divided the population into two main groups, namely, adults and children, for a more accurate risk assessment. Generally, individuals are exposed to harmful pollutants in three ways, including oral intake, inhalation and skin absorption, which can be estimated according to the ”exposure factor manual” [38]. Average daily intake (ADI) is calculated using the following formula.
(7)$${\mathrm{ADI}}_{\mathrm{uptake}}=\frac{{C*\mathrm{IR}}_{s}*\mathrm{EF}*\mathrm{ED}}{\mathrm{BW}*\mathrm{AT}}$$
(8)$${\mathrm{ADI}}_{\mathrm{inhalation}}=\frac{{C*\mathrm{IR}}_{i}*\mathrm{EF}*\mathrm{ED}}{\mathrm{PEF}*\mathrm{BW}*\mathrm{AT}}$$
(9)$${\mathrm{ADI}}_{\mathrm{skin}}=\frac{C*\mathrm{SA}*\mathrm{AF}*\mathrm{ABS}*\mathrm{EF}*\mathrm{ED}}{\mathrm{BW}*\mathrm{AT}}\times 10^{-6}$$
where ADI is the average daily intake (mg/kg-day), C is the concentration of heavy metals in a particular exposure medium (mg/kg), IRs is the uptake rate (kg/day), IRi is the inhalation rate (kg/day), EF is the exposure frequency (day/year), ED is the exposure duration (year), BW is the body weight (kg), and AT is the average dose period (day). For noncarcinogenic effects, AT = ED × 365 (days), for carcinogenic effects, AT = 25,550 (days) (70 years × 365 days/year), SA is the exposed skin surface area (cm^2^), AF is the adhesion factor (mg/cm^2^/day), and ABS is the skin absorption factor (no unit; Appendix A lists the parameter values of the heavy metal exposure assessment model [39,40]. Noncarcinogenic risk was assessed by calculating hazard quotient (HQ) values [41]. For sites contaminated by heavy metal mixtures, the hazard index (HI) was used to summarize the HQ of each heavy metal to assess the overall noncarcinogenic risk. When the HI is lower than the uniform value, it is assumed that the exposed population has no potential risk.
(10)$$\mathrm{HQ}=\frac{\mathrm{ADI}}{\mathrm{RfD}}$$
(11)HI=∑HQ

HI is the estimated noncarcinogenic health risk, and RfD is the reference dose (mg/kg/day) for specific heavy metals [42] (Appendix A). The HQ is used to assess the noncancer health risks of heavy metals in adults and children by ingesting soil in different ways. HQ > 1.0 means that ingestion of soil through ingestion, inhalation or dermal absorption may pose a serious health risk. For carcinogenic risk [43], the carcinogenic risk index of a single heavy metal i is ${\mathrm{Risk}\;}_{i}={\;\mathrm{ADI}}_{\mathrm{Cancer}\;i}\times {\mathrm{SF}}_{i}$, and the comprehensive carcinogenic risk index of all heavy metals is ${\mathrm{Risk}}_{T}={\mathrm{Risk}}_{1}+{\mathrm{Risk}}_{2}+\cdots +{\mathrm{Risk}}_{n}$ (i = 1, 2, 3, …, n), where SFi is the slope coefficient, representing the maximum probability (mg/kg/d) of carcinogenic effects of human exposure to heavy metal i [44,45] (Appendix A); The acceptable levels of ${\mathrm{Risk}\;}_{i}$ and ${\mathrm{Risk}}_{T}$ are between 10^−6^ and 10^−4^, which indicates that there is a certain carcinogenic risk. If it is less than 10^−6^, the carcinogenic risk is not obvious.

## 3. Results

### 3.1. Distribution of Soil Heavy Metals in Mining Areas

The statistical results of soil heavy metal determination in the mining area are shown in Table 2. Compared with the background values of heavy metal elements in Xinjiang soil, the mean values of Zn, Ni, Cu, Cr, As and Cd and in the soil of the Hongshaquan mine drainage site were lower than the background values of Xinjiang soil [23,33]. The coefficients of variation of six heavy metals, Zn, Cu, Pb, As, and Cd, were 16.97%, 16.61%, 12.60%, 17.76%, and 17.16%, respectively, which means they were moderately variable. The variation coefficients of Ni and Cr were 68.55% and 65.89%, which belonged to a high variation, indicating that the spatial distribution of these heavy metals was uneven and discrete. The degree and presence of high value areas were subject to external influences (human activities and coal mining, etc.).

### 3.2. Pollution Factors and Ground Accumulation Index Evaluation Results

Although the average values of soil heavy metals in the study area were mostly lower than the background values, some samples were still higher than the background values. To better evaluate the heavy metal pollution, CF and MCF as well as the ground accumulation index were used to evaluate all samples, and the results are shown in Figure 2. The pollution indices of Zn, Ni, Cu, Cd, As, and Cr in the drainage field and plantation forest areas were less than 1. The CF values of Pb in both areas were between 1 and 2, and some sampling points exceeded 2, indicating that there was slight pollution of Pb elements, while the MCF values in both areas were less than 1, indicating that the drainage field and plantation forest belonged to the nonpollution level.

The results of the ground accumulation index are shown in Figure 3. It is clear that the *I_geo_* values of all elements are less than 0, except for the element Pb, which has a ground accumulation index between 0 and 1 in both areas, and causes slight contamination by element Pb according to the soil ground accumulation index classification criteria (Appendix A).

### 3.3. Potential Ecological Risk Assessment

The potential ecological risk assessment results of soil heavy metal pollution in Xinjiang are shown in Table 3. From Table 3, it can be seen that from the size of the potential ecological risk coefficient $E_{r}^{i}$ of a single heavy metal element, the potential ecological risk coefficient $E_{r}^{i}\;$ values of soil heavy metals in different functional areas are all less than 40, which is classified as a low ecological risk (I). The average values of the potential ecological risk coefficient $E_{r}^{i}$ of the heavy metal element Cd in the dump and plantation soils were 27.37 and 25.16, respectively. The order of the potential ecological risk coefficient of the highest seven heavy metals relative to other elements was Cd > Pb > As > Cu > Ni > Zn > Cr (Figure 4). The potential ecological risk coefficients of Pb, As and Cd were significantly higher than those of the other elements, especially the heavy metal element Cd. Therefore, attention should be paid to the ecological hazards of Pb, As and Cd in mining soil. The total potential ecological risk coefficient RI of soil heavy metals in different functional areas of the mining was less than 150, which belongs to the low ecological risk level (I). Among them, the contribution rate of the heavy metal Cd to the total potential ecological risk was the largest (54.30% and 55.5%), followed by Pb (22.4% and 20.6%), and then As (16.2% and 14.8%). This shows that Pb, As and Cd are the most important risk factors for soil pollution in the mining area.

### 3.4. Health Ecological Risk Assessment

Table 4 shows the noncarcinogenic results of exposure to metals in urban soils through different pathways (ingestion, dermal and inhalation). In terms of noncarcinogenic effects, the noncarcinogenic risk index was higher for children than for adults, which means that children are at a higher health risk due to exposure to metals in mining soils. In addition, the noncarcinogenic risk of all heavy metals showed a characteristic of direct intake > dermal contact > respiratory intake, reflecting the fact that direct intake and dermal contact are the main routes of human exposure to heavy metals in soil. The noncarcinogenic risk and carcinogenicity of elemental Cr in planted forests were higher than those in landfills, suggesting that the abnormal concentration of elemental Cr in planted forests may be due to the high concentration of Cr in a few sites, and the toxicity of Cr directly depends on its valence state [46]; the health risk of Cr therefore needs further study. For noncarcinogenic effects, the HI of all seven heavy metals was less than 1, indicating that the noncarcinogenic risks were within acceptable limits in both areas. In the carcinogenic effect, the carcinogenic risk index of Ni, Pb and As was between 10^−4^ and 10^−6^ (Figure 5), belonging to a low ecological risk, and other elements were almost risk-free. Although the Pb HI values for children are below safe levels (1), landfills have a large amount of traffic, and the main source of Pb is exhaust emissions, which may cause neurological and developmental disorders if children are exposed to sufficiently large doses [47]. Therefore, lead contamination in children should be of concern. In general, the health ecological risks of landfills and plantations are low, but the risks of some trace metals still need to be considered.

### 3.5. Source Analysis of Heavy Metals

To analyze the relationship between heavy metal concentrations, correlation analysis and principal component analysis were performed for each group, while a source identification can be carried out more accurately by multivariate analysis such as PCA (principal components analysis). The results are shown in Table 5. Significant positive correlations (r = 0.689 ** and 0.830 **, respectively) were shown between Zn, Cu, and As in the discharge field, indicating that they have a common source. In addition, the correlation coefficients between Ni and Cu, Cr, Pb, and Cd were all greater than 0.5, and similarly, they may have a common source. The correlation between Cu-Cr-Pb-As was also higher (Figure 6), which may be due to the influence of human activities. For the artificial forest (Figure 6), the correlation coefficients between Zn and Cr and Pb were 0.690 and 0.588, which were significant correlations; Cu-Ni and Cr-As also had significant positive correlations, while Cd had no significant correlation with other elements except for the positive correlation with Cr, which indicated that Cd may have different sources.

According to the above correlation analysis, PCA was performed on the heavy metal data. The results are shown in Table 6. The two principal components were selected in the dump area, and the total contribution rate was 79.41%. In the plantation area, three principal components were selected, with a total contribution rate of 83.76%. Table 6 and Figure 6 show that the contribution rate of the first principal component of the dump is 45.74%, of which the most significant are Cu (0.909), Cr (0.746), Ni (0.743), Pb (0.720), As (0.569), and Zn (0.651), indicating that they have a common source. The contribution rate of component 2 was 33.73%, and only Cd was significant (0.857), indicating that Cd may have a separate source. Cr, Zn, Ni, Cd and Pb are the main components in artificial forests, and they have the same source. Cu (0.953) and As (0.780) were significant in component 2 and component 3, respectively, indicating that these two elements have different sources.

## 4. Discussion

In this study, the pollution characteristics of Zn, Ni, Cu, Cr, Pb, As and Cd in different functional areas of open pit mines were analyzed. The results show that the Pb element in the two functional areas is the main pollution source that causes the ecological risk in the mining area, which is inseparable from the traffic activities of the mining area, because Pb is mainly derived from the emission of automobile exhausts. The potential ecological risk and human health risk assessment showed that the overall ecological risk of the mining area was low, and the noncarcinogenic risk index (RI) was lower than 1. Among them, the noncarcinogenic risk to children and adults was different, mainly because the intake rate of adults was higher than that of children, while the inhalation rate of children was higher than that of adults, and the respiratory system of children was more fragile, which easily caused health risks [48,49,50]. Although the seven heavy metals have no noncarcinogenic risk, the risk of heavy metal elements to children still needs attention. Regarding the carcinogenic risk index, the hazards of Ni, Pb and As are worthy of attention. Pb and As are still at a tolerable level, but the carcinogenic risk of Ni is higher, close to 10^−4^, which should be considered. To prevent the health hazards to the mining population, it is necessary to strengthen the protection measures for it.

PCA showed that the contribution rate of principal component 1 of the dump was 45.74%, and the main contributing elements were Cu, Cr, Ni, Pb, As and Zn. Among them, Cr, Zn, Ni and Cu are derived from the wear on tires and other automobile parts [51], and Pb is caused by automobile exhaust pollution [52]. As is caused by fuel combustion and atmospheric deposition from nearby factories [53]; it is identified as both a traffic source and an industrial source. Therefore, they were identified as traffic sources and industrial sources. The load of Cd in component 2 is higher, and its concentration is similar to the local soil background value, mainly from the soil parent material. The contribution rate of principal component 1 to the artificial forest was 46.63%, and the main elements were Cr, Zn, Ni, Cd and Pb related to vehicle activities [54,55], which were identified as traffic sources, while the main element of principal component 2 was Cu. This may be because the water used for artificial forest irrigation is wastewater from the treated mine [56]. Component 2 is an industrial source. The load of As in the third component is the highest, but it does not exceed the soil background value, mainly from the soil parent material [57,58]. Overall, the impact of human activities on the dump is more serious. This may be due to artificial forests in the main road of the mining area and tree dust, reducing the impact of heavy metals carried by the atmosphere. The dump is next to the mine, and transport vehicles are more frequent, operate on a high terrain, have no vegetation protection, and are more vulnerable to dust and cause other factors.

In general, there were some uncertainties in this study. The complex terrain of the mining area limits the sampling points. The sampling points are concentrated in two special areas, which may not represent the heavy metal pollution of the entire mining area. Future research will expand the sampling area and the number of sampling points, making the research results more scientific.

## 5. Conclusions

1. The soil pH in the study area is mainly alkaline soil; only the average content of Pb in heavy metals exceeds the local background value of Xinjiang, and the average content of other elements does not exceed the background value. According to the pollution index method and the geoaccumulation index method, except for the slight pollution of Pb, other heavy metal elements are essentially pollution-free. The potential ecological risk index method shows that both the dump and the plantation area belong to a low ecological risk.

2. The human health risk assessment showed that, for children and adults, the main exposure pathways were ingestion and dermal contact, and the contribution of Pb and As to noncarcinogenic risk was higher. The noncarcinogenic risk of Cr in the artificial forest area was significantly higher than that in the dump, which may be caused by the higher content of Cr in the soil parent material. For the noncarcinogenic risk, neither region produces risk. For the carcinogenic risk, except for Ni, Pb, and As, which carry a lower carcinogenic risk, the remaining heavy metals carry no carcinogenic risk. It is worth noting that adults run a higher risk of cancer than children, which may be due to prolonged skin contact.

3. The results of the PCA source analysis showed that the sources of heavy metals in the dump can be divided into two categories. Factor 1 is mainly Cu, Cr, Ni, Pb, As, Zn, and their sources are related to transportation and industrial activities, linked to human sources. The contribution of Cd in factor 2 is larger, which is related to the soil parent material and is linked to natural sources. In the artificial forest, the loads of Cr, Zn, Ni and Pb in factor 1 are larger; they mainly come from traffic, and the contribution of Cu in factor 2 is the highest; this comes from industrial sources. The As in factor 3 is related to soil parent material. Compared with the other two, the dump is more disturbed by human factors than the plantation.

## Figures and Tables

**Figure 1 ijerph-19-15522-f001:**
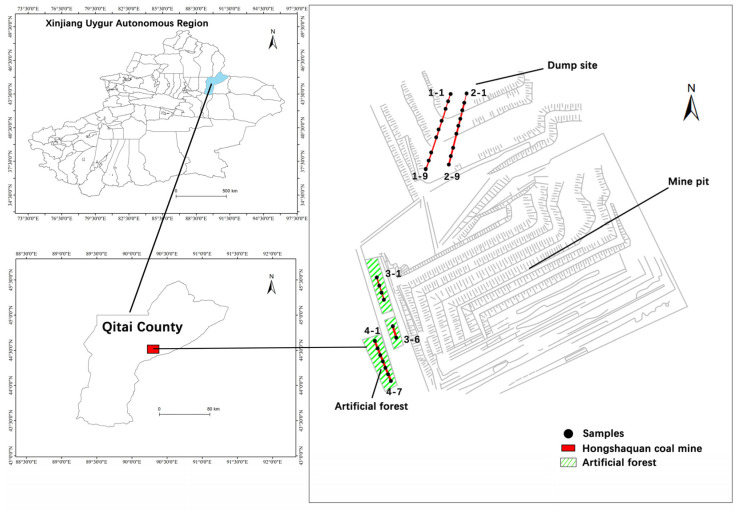
Sampling point layout map.

**Figure 2 ijerph-19-15522-f002:**
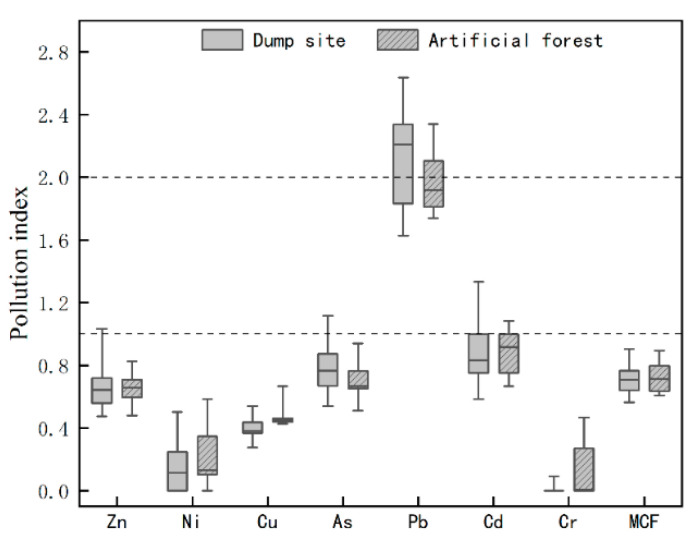
Boxplot of Heavy Metal Pollution Index.

**Figure 3 ijerph-19-15522-f003:**
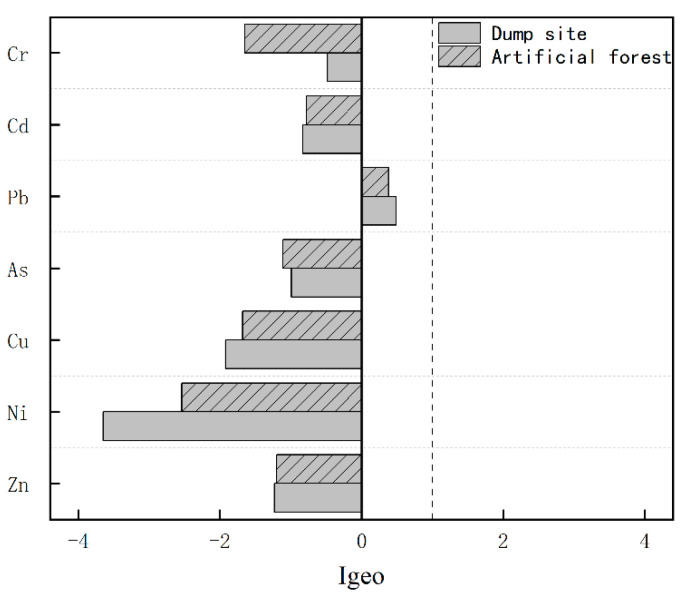
Distribution diagram of ground accumulation index.

**Figure 4 ijerph-19-15522-f004:**
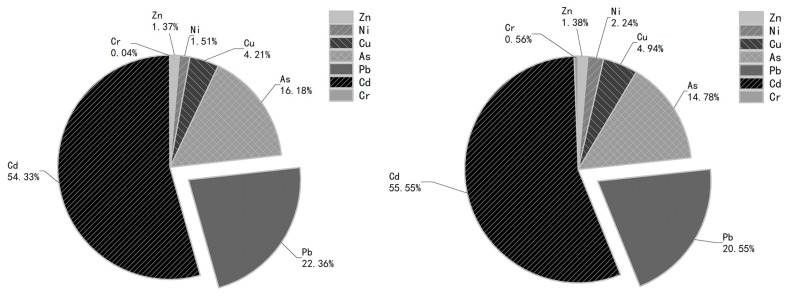
Potential risk contribution diagram of heavy metals (**left** for dump site, **right** for artificial forest).

**Figure 5 ijerph-19-15522-f005:**
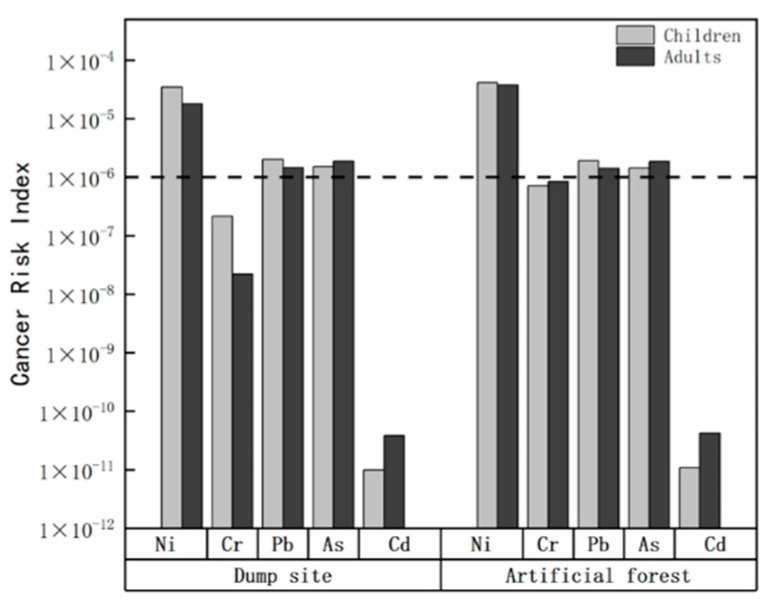
Cancer risk index of different functional areas.

**Figure 6 ijerph-19-15522-f006:**
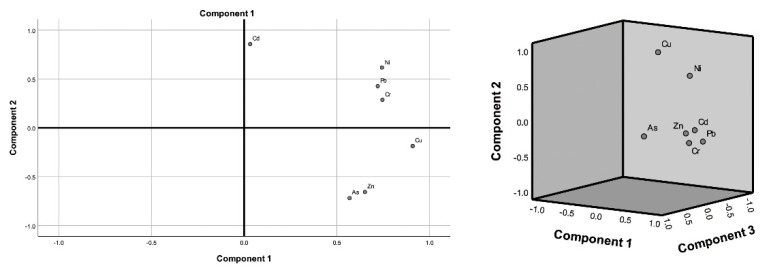
Heavy metal correlation analysis (**left** for dump, **right** for artificial forest).

**Table 1 ijerph-19-15522-t001:** Number and location of sampling points.

Area	Number of Points	Numbering	Longitude & Latitude	
			Longitude	Latitude
Dump site	18	1-1–1-9	E 90°16′40″	N 44°30′42″
		2-1–2-9		
Artificial forest	13	3-1–3-6	E 90°16′50″	N 44°30′10″
		4-1–4-7		

**Table 2 ijerph-19-15522-t002:** Statistical characterization of heavy metal content of soils in drainage sites (mg/kg).

	Zn	Ni	Cu	Cr	Pb	As	Cd
Maximum	71.12	15.43	23.87	18.43	34.09	10.14	0.16
Minimum	32.70	0.06	9.84	0.10	21.98	4.64	0.07
SD	7.64	4.25	2.56	5.49	3.50	1.20	0.02
Average	45.04	6.19	15.40	8.33	27.78	6.75	0.10
CV (%)	16.97	68.55	16.61	65.89	12.60	17.76	17.16
Soil background in Xinjiang, China	68.80	26.40	35.80	39.60	13.50	9.09	0.12
China Standard Value (pH > 7.5)	300	190	100	250	170	25	0.6
Exceeding the background value ratio (%)	0	0	0	0	100	3.23	16.7

**Table 3 ijerph-19-15522-t003:** Potential ecological risk index of heavy metals in different regions.

Area	Potential Ecological Risk Coefficient of Single Element $E_{r}^{i}$							Comprehensive Potential Ecological Risk Coefficient RI
	Zn	Ni	Cu	As	Pb	Cd	Cr	
Dump site	0.65	0.72	2	7.69	10.63	25.83	0.02	47.55
Artificial forest	0.66	1.07	2.36	7.06	9.82	26.54	0.27	47.78

**Table 4 ijerph-19-15522-t004:** Soil heavy metal non-carcinogenic risk index.

	HQ_skin_		HQ_inhalation_		HQ_uptake_		Non-Cancer Risk Index (Hi)	
	Children	Adults	Children	Adults	Children	Adults	Children	Adults
Dump site								
Zn	1.19 × 10^−6^	2.24 × 10^−5^	2.37 × 10^−9^	9.25 × 10^−9^	2.12 × 10^−5^	1.57 × 10^−5^	2.24 × 10^−5^	3.81 × 10^−5^
Ni	1.68 × 10^−6^	2.18 × 10^−5^	1.01 × 10^−6^	2.70 × 10^−6^	4.06 × 10^−5^	2.07 × 10^−5^	4.33 × 10^−5^	4.52 × 10^−5^
Cu	1.89 × 10^−6^	3.73 × 10^−5^	1.58 × 10^−8^	6.46 × 10^−8^	5.07 × 10^−5^	3.93 × 10^−5^	5.26 × 10^−5^	7.67 × 10^−5^
Cr	7.91 × 10^−5^	1.99 × 10^−4^	1.66 × 10^−6^	8.63 × 10^−7^	1.41 × 10^−4^	1.40 × 10^−5^	2.22 × 10^−4^	2.14 × 10^−4^
Pb	8.68 × 10^−5^	1.58 × 10^−3^	1.30 × 10^−7^	4.90 × 10^−7^	1.16 × 10^−3^	8.33 × 10^−4^	1.25 × 10^−3^	2.42 × 10^−3^
As	9.02 × 10^−5^	1.70 × 10^−3^			3.30 × 10^−3^	2.45 × 10^−3^	3.39 × 10^−3^	4.15 × 10^−3^
Cd	1.59 × 10^−5^	2.93 × 10^−4^	1.58 × 10^−7^	6.05 × 10^−7^	1.42 × 10^−5^	1.03 × 10^−5^	3.02 × 10^−5^	3.04 × 10^−4^
Artificial forest								
Zn	1.20 × 10^−6^	2.19 × 10^−5^	2.40 × 10^−9^	9.03 × 10^−9^	2.14 × 10^−5^	1.53 × 10^−5^	2.26 × 10^−5^	3.72 × 10^−5^
Ni	1.98 × 10^−6^	4.57 × 10^−5^	1.19 × 10^−6^	5.66 × 10^−6^	4.77 × 10^−5^	4.33 × 10^−5^	5.09 × 10^−5^	9.47 × 10^−5^
Cu	2.26 × 10^−6^	3.98 × 10^−5^	1.89 × 10^−8^	6.90 × 10^−8^	6.05 × 10^−5^	4.19 × 10^−5^	6.27 × 10^−5^	8.18 × 10^−5^
Cr	2.64 × 10^−4^	7.64 × 10^−3^	5.52 × 10^−6^	3.31 × 10^−5^	4.71 × 10^−4^	5.36 × 10^−4^	7.40 × 10^−4^	8.21 × 10^−3^
Pb	8.13 × 10^−5^	1.52 × 10^−3^	1.22 × 10^−7^	4.71 × 10^−7^	1.09 × 10^−3^	8.01 × 10^−4^	1.17 × 10^−3^	2.32 × 10^−3^
As	8.43 × 10^−5^	1.67 × 10^−3^			3.08 × 10^−3^	2.41 × 10^−3^	3.17 × 10^−3^	4.08 × 10^−3^
Cd	1.72 × 10^−5^	3.26 × 10^−4^	1.71 × 10^−7^	6.72 × 10^−7^	1.53 × 10^−5^	1.14 × 10^−5^	3.27 × 10^−5^	3.38 × 10^−4^

**Table 5 ijerph-19-15522-t005:** Correlation Analysis.

Area	Zn	Ni	Cu	Cr	Pb	As	Cd
Dump site							
Zn	1						
Ni	0.074	1					
Cu	0.689 **	0.537 *	1				
Cr	0.176	0.740 **	0.653 **	1			
Pb	0.227	0.758 **	0.473 *	0.438	1		
As	0.830 **	−0.026	0.580 *	0.116	0.184	1	
Cd	−0.421	0.507 *	−0.122	0.079	0.416	−0.492 *	1
Artificial forest							
Zn	1						
Ni	0.427	1					
Cu	0.109	0.772 **	1				
Cr	0.690 **	0.487	−0.029	1			
Pb	0.588 *	0.277	−0.109	0.544	1		
As	0.451	0.264	0.025	0.624 *	0.013	1	
Cd	0.284	0.494	−0.03	0.637 *	0.463	0.236	1

* *p* < 0.05, ** *p* < 0.01.

**Table 6 ijerph-19-15522-t006:** Principal component analysis.

Dump Site	1	2	Artificial Forest	1	2	3
Cu	0.909	−0.187	Cr	**0.904**	−0.243	0.156
Cr	**0.746**	0.286	Zn	**0.793**	−0.129	0.073
Ni	**0.743**	0.617	Ni	**0.724**	0.654	−0.120
Pb	**0.720**	0.426	Cd	**0.700**	−0.144	−0.275
Cd	0.032	**0.857**	Pb	**0.647**	−0.351	−0.551
As	**0.569**	−0.719	Cu	0.239	**0.953**	−0.037
Zn	**0.651**	−0.657	As	0.571	−0.092	**0.780**
Total	3.202	2.361	Total	3.264	1.565	1.033
Variance (%)	45.74	33.73	Variance (%)	46.62	22.35	14.75
Cumulative (%)	45.74	79.47	Cumulative (%)	46.62	68.98	83.74

Boldface font indicates a higher value of each component load.

## Data Availability

Not applicable.

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
