# Peer review of "Ecological Risk Assessment and Source Apportionment of Heavy Metals in the Soil of an Opencast Mine in Xinjiang"

_ijerph, 2022, doi:10.3390/ijerph192315522_

Round 1
Reviewer 1 Report
In the study, the effects caused by the open pit mine were examined using a number of indices. First of all, nice work. The presentation of the article is self explanatory. But I have a few concerns about the article.
-You did not mention the PCA expansion in the abstract and the rest of the article. Only the abbreviation is used. You need to add it.
-You should include more studies on this subject in the literature. I would like to see more references to the studies on the indexes you use.
-Also, the purpose of the study should be emphasized more. Before line 57, you need to explain why the work was done. Just for reference?
- How 31 samples were taken. Is ^1 sample sample sufficient to perform such an analysis? What are the sampling density of the samples, the soil depth, the distances between the samples? There are ambiguities in this section and should be clearly explained. For example, in Table 1, 2-1-7, followed by the number denotes the sample numbers. What is 2-1? You need to add a description about the way the samples are received and displayed.
- In Figure 5, there is no cd value in children, what is the reason for this?
-Why did you get the samples from dump site and artifical forest?
I think the article will be more understandable if you include the answers to these questions in the article
Reviewer 2 Report
Overall, the presented manuscript is very interesting, but there are a lot of inconsistencies and lot of typos and grammar errors.
There are also some unacceptable errors
Part soil sample collection does not contain important information about sample processing, e.g. were samples analysed in triplicate or only once? Was open digestion used? Or did the authors use digestion in closed system?
e.g. line 185-186 the authors reported coefficients of variation, but the data in table 8 (CV) are different. In the text for Zn is value 26 % but in table 8 the CV is 17 %,…..
Line 13 „… contents of 7 heavy metals (As, Cd, Cr, Cu, H, Ni, Pb and Zn) and pH…..“ There is 8 symbols of element. Is H hydrogen?
Line 32 „… reported to be 16.1 % in 2014%....“ typpo 2014%
Reviewer 3 Report
This paper describes the Ecological Risk Assessment and Source Apportionment of Heavy Metals in the Soil of an Opencast Mine in Xinjiang. After reviewing it, I think it can be consider to publish if the following issues are solved:
1. For the introduction part, more disccussion about the heavy metals in soil should be added such as : what kind of metals.
2. The tables in the manuscript can be reduced.
3. Some relevant papers should be cited: Small Methods 2200314 (2022) doi:10.1002/smtd.202200314; Colloids and Surfaces A: Physicochemical and Engineering Aspects 646, 128962; Separation and Purification Technology 303 (2022) 122288.
Round 2
Reviewer 2 Report
In the abstract is still: The contents of 7 heavy metals (As, Cd, Cr, Cu, H, Ni, Pb and Zn) in the soil were analyzed. I really don't understand what the symbol H means. Is it a symbol for hydrogen. Did authors really measure 7 heavy metals and hydrogen. Please explain it.
Author Response
请参阅附件。
